# Genetic Diversity and Population Structure for Resistance and Susceptibility to Mastitis in Braunvieh Cattle

**DOI:** 10.3390/vetsci8120329

**Published:** 2021-12-14

**Authors:** Mitzilin Zuleica Trujano-Chavez, Reyna Sánchez-Ramos, Paulino Pérez-Rodríguez, Agustín Ruíz-Flores

**Affiliations:** 1Posgrado en Producción Animal, Universidad Autónoma Chapingo, Carretera Federal México-Texcoco Km 38.5, Texcoco 56227, Estado de México, Mexico; zulealizee@gmail.com; 2Recursos Genéticos y Productividad, Colegio de Postgraduados, Carretera Federal México-Texcoco Km 36.5, Texcoco 56230, Estado de México, Mexico; rey_1014@hotmail.com; 3Socio Economía Estadística e Informática-Estadística, Colegio de Postgraduados, Carretera Federal México-Texcoco Km 36.5, Texcoco 56230, Estado de México, Mexico; perpdgo@gmail.com

**Keywords:** inbreeding, heterozygosity, candidate genes, principal components analysis

## Abstract

Mastitis is a disease that causes significant economic losses, since resistance to mastitis is a difficult trait to be improved due to its multifactorial occurrence. Therefore, our objective was to characterize a Mexican Braunvieh cattle population for genetic resistance and susceptibility to mastitis. We used 66 SNP markers for 45 candidate genes in 150 animals. The average heterozygosity was 0.445 ± 0.076, a value higher than those reported for some European breeds. The inbreeding coefficient was slightly negative for resistance to subclinical (−0.058 ± 0.055) and clinical (−0.034 ± 0.076) mastitis, possibly due to low selection for the immunological candidate genes that influence these traits. The genotypic profiles for the candidate loci per K-means group were obtained, as well as the group distribution through the graphics of the principal component analysis. The genotypic profiles showed high genetic diversity among groups. Resistance to clinical mastitis had the lowest presence of the heterozygous genotypes. Although the percentage of highly inbred animals (>50%) is up to 13.3%, there are highly heterozygous groups in terms of the studied traits, a favorable indicator of the presence of genetic diversity. The results of this study constitute evidence of the genetic potential of the Mexican Braunvieh population to improve mastitis-related traits.

## 1. Introduction

One of the major problems in dairy units is mastitis [1]. Tropical dual-purpose production systems have the same problem. A prevalence of 60% was reported for mastitis in tropical production units [2]. The current trend seems to focus on milk quality rather than on quantity, and mastitis has been one of the main hurdles [1,3].

Some of the obstacles that traditional genetic selection has faced in bovines are as follows [4]: (1) complexity of quantitative or low-heritability traits, (2) traits measured late in the life of the animals, and (3) traits that are difficult to be measured such as resistance and susceptibility to mastitis. In particular, to reduce the limitation of the complexity of quantitative traits, an attempt has been made to improve the animals by detecting quantitative trait locus (QTL) in the genomes. A QTL is a section of DNA that is correlated with a quantitative trait or phenotype of a population. Traits of economic importance in animals are polygenic, that is, there are several QTLs that explain the phenotypic variation [4]. These QTLs indicate the genetic architecture of the traits, helping to find candidate genes.

Resistance to mastitis is a complex trait that is multifactorial in nature and, thus, difficult to be improved. The genomic prediction ability for clinical mastitis is low, 0.19 [5], compared with that reported for residual feed intake (0.85 ± 0.08), dry matter intake (0.52 ± 0.04), and feed conversion ratio (0.46 ± 0.05) [6]. The conventional selection scheme for resistance has not been effective against mastitis due to the large number of candidate genes or QTLs related to non-immune factors that influence mastitis and the factors that make resistance to mastitis measurement difficult [4,7,8].

In addition to the control of environmental factors to prevent mastitis at the farm level, genomic methods have been used as an alternative solution to the problem of increasing resistance to mastitis. The aim is to obtain animals that are immune to the main mastitis-causing pathogens, and that have qualities that allow them to adapt to environmental factors that cause mastitis [2].

Recently, the genotyping of animals has become easy and relatively cheap. A sample of hair or blood will provide thousands of markers [4]. The markers mostly used for genomic selection are single nucleotide polymorphism (SNP), which are a type of polymorphism that produces a variation in a single base pair. Their generalized use has reduced the cost of genotyping to a cost-effective level. Genotyping is widespread and is used in both commercial and research systems [9].

Manipulation of genetic variation is key to the development and use of any genetic improvement program for dairy and beef cattle breeds [10]. Genetic diversity has been studied in original Braunvieh and derivative breeds, with the aim of disentangling the genetic basis of their adaptation to diverse environments and increasing the productive performance. The results have been positive. It was found that populations of this breed adapted to environments different from that of its origin are genetically more diverse than the average of other *Bos taurus* and some *Bos indicus* breeds [11]. Animals living in harsh environmental conditions, such as extreme temperatures, possess greater genetic diversity, as facing environmental changes require versatility [12].

Genetic diversity is determined by population parameters and visual dimension reduction tools. The Hardy–Weinberg equilibrium, the coefficient of consanguinity and heterozygosity are the parameters most used in genetic diversity studies. The Hardy–Weinberg equilibrium states that the allele and genotype frequencies in a population will remain constant from generation to generation in the absence of assortative mating, natural selection, gene flow, mutation, and inbreeding. On the other hand, principal component analysis (PCA) has been proven to be useful to visualize the genotypic distribution of animals [11,13].

The objective of this study was to determine the genetic diversity and population structure of a Mexican Braunvieh population in terms of mastitis resistance and susceptibility, using (1) basic population statistics: inbreeding coefficient, expected and observed heterozygosity, and (2) multivariate statistics, namely K-means and principal components analysis.

## 2. Materials and Methods

### 2.1. Source of Information

Genomic information was obtained using hair samples from 150 animals born between 2001 and 2016 on five farms belonging to the Mexican Association of Braunvieh Purebred Breeders in Eastern, Central, and Western Mexico. The samples were genotyped at GeneSeek (Lincoln, NE, USA, http://neogene.com, accessed on 28 May 2021). The chip used was the Genomic Profile Bovine Low-Density with 50,000 SNP markers.

### 2.2. Genotype Quality Control

For data edition, quality control was performed prior to the genotypic analyses. The missing genotypes were imputed using the observed allele frequencies. The imputation method consisted of assigning an allele, according to the probability of a possible type of polymorphism for a certain marker in the entire population. Additionally, a 0.05 threshold for the minor allele frequency was considered to discard the non-informative markers or those with errors in the genotyping.

### 2.3. Statistical Analyses

#### 2.3.1. Identification of Associated and Informative Loci

The candidate genes associated with mastitis in *Bos taurus* were identified, according to reports by [14,15,16,17] for resistance to mastitis (RM), by [18,19] for susceptibility to mastitis (SM), by [20,21,22] for resistance to bacterial mastitis (RBM), by [8] for resistance to subclinical mastitis (RSM), and by [23] for resistance to clinical mastitis (RCM).

Candidate genes reported for RM are involved with risk factors such as pathogens, udder structure, age, transition period, host nutritional stress, and the immune system [24]. RM comprises three other traits of this study: RBM, RSM and RCM. We tried to determine the differences in genetic diversity and population structure when studying resistance to different types of mastitis. The candidate genes for RBM, RSM, and RCM are different from those reported for RM. For example, in RBM, the pathogenic factor has a greater impact, so the candidate genes for the characteristic are particularly associated with immunological processes and the production of antibodies [14,24]. Finally, for the study, SM was considered, where the loci under study are not the same as for RM, even though they are opposite, as candidate genes for SM are more associated with the immune system response than with the other risk factors mentioned for RM [24,25].

The positions, ranked by base pairs, were obtained from the Gene Library [26]. With this information, the intragenic loci of the candidate genes were searched; the criterion for the loci-gene association was that they should be located within the position rank ± 25 kb of the gene for the same chromosome. Only the loci available in the genotypic database were included. The analyses considered 455 SNP markers of 204 candidate genes reported in the literature associated with resistance and susceptibility to mastitis.

All of the analyses were carried out using R software, version 4.0.4 [27]. According to the literature consulted, the 10% most informative markers associated with each of the studied traits were determined using the Shannon index. This index allows for the identification of the most informative markers [28]. The index is defined as:(1)$$H=-\overset{2}{\underset{i=0}{\sum }}p_{i}\;log\;p_{i},$$
where *i* = 0, 1, 2 refers to the genotype (AA = 0, AB = 1, BB = 2) and *p_i_* is the corresponding allelic frequency. This index was calculated for each of the loci. Empirical distribution was obtained, and the 10% most informative markers were selected. The highest values for H were associated with the highest diversity for the marker. At the end of filtering, 66 SNP markers of 45 candidate genes were left for further analyses. For better handling of the markers in the graphs, the names of the loci were replaced by consecutive numbers, assigned by alphabetical order to the corresponding candidate gene.

#### 2.3.2. Population Genetic Structure

The expected (He) and observed (Ho) heterozygosity were estimated for all the loci by trait in the study using the adegenet package [29] available in R [27]. The stats package of R [27] was used to perform the *t*-test to determine the differences between He and Ho. The null hypothesis was that there were non-significant differences between them. A *p*-value of 0.01 was used to declare significance and was corrected with the Bonferroni method considering the total number of loci by trait.

The pegas package [30] through adegenet [29] was used to determine whether the loci under study were in the Hardy–Weinberg equilibrium. A hundred thousand replicates were assigned to execute the test procedure through the Monte Carlo method. The null hypothesis established the existence of the Hardy–Weinberg equilibrium when *p*-values were higher than 0.01/*n* with the Bonferroni correction; *n* is the number of loci per trait.

The inbreeding coefficients by individual with respect to the population (F_IT_) were calculated and plotted with adegenet [29]. Additionally, the general mean for F_IT_ per trait was obtained, along with its respective standard deviation. Finally, a histogram including a nonparametric estimate of the density function by trait was obtained. This was done to visualize the animals with an inbreeding coefficient above 0.5 (the number of animals could be observed in the histogram).

#### 2.3.3. Cluster Analyses

The differences in the genotypic profile of individuals were identified with two clustering algorithms: hierarchical clustering using the Ward distance implemented in the stats package and the K-means algorithm also included in the stats package in R and factoextra [31]. For both methods, the 10% of the markers, identified in a previous step, was used.

The optimum number of groups (*k*) to be formed through the hierarchical clustering and K-means was obtained with Silhouette and Elbow of factoextra [31]. A search for *k* that allowed for an equilibrium between a value of the silhouette close to one [32] and the least error sum of squares was carried out.

The Ward algorithm of minimum variance, according to [33], is a divisive hierarchical method that creates mutually exclusive sub-sets. Sub-set members are ultimately similar to each other and different between groups. This promises to be a good method for finding different groups with genotypic profiles. The results of the procedure are illustrated in a circular dendrogram that displays the hierarchy of the groups and their similarity to the groups formed with the K-means algorithm.

K-means is a non-hierarchical algorithm that uses a reassignment method based on centroids to form groups. This is the most frequently used method to form groups [34]. It is used in robotics, genomics, and genetic diversity. In our study, the groups of K-means were used to identify divergent genotypes among groups of animals and to characterize genetic diversity based on the PCA.

#### 2.3.4. Principal Components Analysis

The PCA was performed using adegenet package [29] through the sample covariance matrix obtained using the candidate SNP. The objective was to observe the distribution of the groups formed by K-means through a PCA graphic of individuals (first and second dimensions) and to reduce the number of markers according to their eigenvalues. With a factoextra biplot [31], we estimated the correlation and contribution of markers to the first and second dimensions formed by the PCA and determined the 10% of animals with the highest contribution to the analysis.

## 3. Results and Discussion

### 3.1. Sample Size and Power Analysis

In México, studies using genomic information from large animal populations are scarce. Usually, the main restriction is the availability of economic resources for animal genotyping. Genetic improvement infrastructure of developing countries usually is less robust than that of developed countries, whose genomic databases involve millions of animals [35].

In previous studies carried out with the same population used in our study, the genomic information of 300 animals was considered to find candidate markers for diseases [36] and meat quality [37]. For this study, we used intragenic markers (±25 kb), and only 150 animals with candidate genes associated with resistance and susceptibility to mastitis were found.

The sample size used in our study, evaluated with the pwr.chisq.test function of the pwr package [38], was enough to detect significant differences with a *p*-value of 0.05 and an effective size of 0.4. The power of 0.87 for the Xi^2^ test obtained in the present study is adequate to obtain conclusive, reliable results [39]. In other studies of genetic diversity and structure, smaller sample sizes have been used, and conclusive results have been reached: 6 to 23 [40], 7 to 58 [41], 13 to 38 [42], and 71 to 167 [43].

### 3.2. Identification of Associated and Informative Loci

Table 1, Table 2, Table 3, Table 4 and Table 5 show the markers obtained after filtering with the Shannon Index. Their candidate genes, chromosome, and the number assigned to visualize the results in the graphs are shown. For RM, 19 genes with 22 loci were obtained; for SM, 7 genes and 11 loci; for RBM, 7 genes and 8 markers; for RSM, 8 genes and 20 markers; and for RCM, 4 genes and 6 markers. This adds up to 66 markers and 45 candidate genes. All loci were in the Hardy–Weinberg equilibrium.

### 3.3. Population Genetic Structure

Table 6 shows the results for He and Ho, as well as those of a *t*-test to determine whether the differences were statistically significant. The rule of decision to not reject the null hypothesis of non-significant differences between He and Ho was that the *p*-value for the *t*-test > *p*-value for Bonferroni Correction. In the last column, the |He-Ho| differences can be observed; all were non-significant, with exception of the difference for SM.

The estimates for Ho obtained in our study for mastitis (average 0.45 ± 0.076) are above those obtained with breeds derived from the original Braunvieh, but below those reported for *Bos taurus* breeds, with some adaptation to the tropics. In Colombia, Creole cattle [44] reported Ho of 0.66, and the Costeño con Cuernos breed had the lowest, 0.635, and Casareño the highest, 0.733. These Ho estimates are quite different from those reported by [45], who found an average of 0.35 ± 0.167 for Ho in populations related to Braunvieh.

This suggests that the Mexican Braunvieh population has increased its genetic versatility for the candidate genes studied, with a higher number of heterozygous individuals. Exposure to climates different from their native climate for over a century in the tropical systems of Mexico could have caused a change in Ho, with respect to original Braunvieh populations [13,45]. On the other hand, the non-significant differences between He and Ho for most of the traits (Table 6) is similar to the results obtained in other studies, where neither the breed nor the environment modified the non-rejection of the null hypothesis, He = Ho for Xi2 [13,44,46].

As a graphic resource, Figure 1 shows that the absolute differences at the locus level are small, as in the overall result. The exception to this was SM, whose joint trend of the loci was to be over 0.03. Thus, it is the only trait with significant differences. In the bar-plots for the rest of the traits, there were loci that were both close to zero and above 0.05, but the overall estimate is not enough to declare evidence for a significant difference.

The estimate for the F_IT_ obtained coincides with those observed in semi-specialized systems of production. The overall average for F_IT_ was 0.017 ± 0.043 for all the traits associated with mastitis. This value is similar to those found for Sahiwal (0.013 ± 0.109), Gyr (0.013 ± 0.106), and Guernsey (0.02 ± 0.217) [47]. In contrast, the F_IT_ for the tropical breeds Landim, Angole, and Tete [48] was 70% higher than that found in our study. Furthermore, for Creole Colombian breeds, the average was 0.09 [44]. The first group of breeds was maintained in semi-specialized systems [47], while Landim, Angole, Tete, and the Colombian Creole are non-specialized breeds. This difference could explain why smaller breeds without adequate genetic control present higher inbreeding coefficients.

Figure 2 shows that, for traits RM, SM, and RBM, most of the loci had positive F_IT_. In contrast, the loci for RSM and RCM were mostly negative; that is, there were no traces of inbreeding. This could be due to the specificity of the last two traits, where the candidate genes involve DNA segments that have undergone null direct selection.

The genes whose symbol starts with CXCL, chemokine (C-X-C motif) ligand, are responsible for the immune response [49]. An in vivo study demonstrated that the CXCL genes are key in the inflammatory process of the mammary gland, after the entry of bacteria through the cow’s nipple, particularly the CXCL8 and CXCL10 genes [50,51]. In our study, for RCM, 67% of the SNP markers belong to genes CXCL1 and CXCL8. This explains the low F_IT_ values for the trait, which is because current improvement programs for the Braunvieh population under study have no selection criteria based on resistance to mastitis.

For RSM, 40% of the markers belong to EDN2 and HIVEP3, candidate genes associated with immune processes, such as intracellular and sequential signaling of immunoglobulin receptors. Because of their immunologic nature, these genes have not been directly selected for; however, at the farm level there is a rising concern for improving their animals for genes associated with diseases that could be prevented, such as paratuberculosis and clinic mastitis [8,52].

The distribution of animals by their F_IT_ value can be observed in Figure 3. In general, animals fall in the range of 0 to 0.3 F_IT_. F_IT_ = 50% is equivalent to the value of an inbred animal produced by two progenitors with 100% genetic relationship. Our animals varied from 1.3% to 13.3%. The trait with the highest number of highly inbred animals was SM, while RSM had the lowest.

Although up to 13.3% of animals have non-desirable F_IT_ values, the estimate obtained in our study (0.017 ± 0.043) is similar to that reported for other cattle breeds in specialized and semi-specialized systems of production. In those systems, although some animals had high inbreeding coefficients, the average F_IT_ is lower than those reported by [53] for *Bos taurus* breeds (0.071), Brown Swiss (0.071), Braunvieh (0.059), Original Braunvieh (0.023), Holstein (0.057), and Simmental (0.028).

### 3.4. Clustering Using Hierarchical Methods and K-Means Algorithms

The optimum number of *k* to equilibrate the silhouette width and the means of squared errors for both algorithms converged at the same number of clusters for each trait. However, better values for silhouette were found by grouping with the K-means algorithm: RM, 0.08; SM, 0.17; RBM, 0.17; RSM, 0.12; and RCM, 0.39. For the first four traits, the silhouette values were low, but in all cases, they were negative, a good indicator of the suitability of the method. The RCM silhouette was the widest.

The average proportion of variability among groups relative to the total variability was obtained. The estimates found were RM, 34.9%; SM, 65.5%; RBM, 66.9%; RSM, 54.7%; and RCM, 77.3%. It should be noted that the RCM values for variability and silhouette were the best of all the traits. This could be due to the small number of markers of the trait (6), which were highly informative as well.

The circular dendrograms of Figure 4 were generated to visualize the groups formed by the Ward hierarchical method. The groups created with this algorithm were very similar to those created with K-means. The *k* number, the number of individuals per group, and the representative animals together with their genotypic profile can be seen in Figure 5. The representative animal is the animal closest to the centroid of its cluster, and its genotypic profile was obtained to visualize the main differences among group patterns.

Ho and F_IT_ are not related because, unlike inbreeding, heterozygosity is not correlated with the markers [54], even though Ho values close to 0.50 were found for all the traits. In the genotypic profiles, it can be observed that there are groups of animals with a low frequency of genotype AB (Figure 5). The traits with the lowest values for F_IT_ present high numbers of homozygous animals. This could be explained by the low genetic relationship among the animals for these loci, that is, genotypes AA or BB for the same locus, as seen in Table 6. One of these traits (SM) presents a significant He-Ho difference. This means a decrease in the expected value for heterozygosity, which will be reflected in a greater number of homozygous individuals.

The trait with the greatest number of heterozygous animals was RM. Groups III, VII, VIII, and XI possess most of their loci with genotype AB. RM is multifactorial, and therefore, it includes candidate genes associated with productive, reproductive, adaptative, and immunological traits. This will cause high genetic diversity, in contrast with the traits RSM and RCM, which, in our study, are only influenced by immunological candidate genes of a specific order whose diversity is limited. According to [55], these gene types tend to be found in homozygous genotypes in wildlife, given that AB genotypes are associated with proteinic abnormalities that increase their susceptibility to pathogenic diseases.

For SM, there are well defined groups of animals whose loci are mostly heterozygous. Groups II, IV, V, XV, and XVII have at least 55% of their loci with genotype AB. The groups with fewest heterozygous individuals were III, XVIII, XIX, and XX, with up to 18% AB genotypes. There is high genetic diversity in the population for the trait as was observed for RM; the reason could be the presence of genes influencing traits that are indirectly related with mastitis, and not just loci affecting the immune system [49,55].

The loci of the groups formed by the algorithm K-means for RBM are from 29% (groups II, VI, IX, and XX) to 57% (groups I, III, VIII, and XII) heterozygous. Diversity was high because only 33% of the genes were directly related with immune functions in the animal, genes IL1A and IL1RN, while the rest were related to metabolic processes [20,21,22].

### 3.5. Principal Components Analysis (PCA)

The PCA was useful to observe the genetic diversity in the groups of animals generated with K-means. The first two dimensions of the PCA explained 22.21% (RM), 42.1% (SM), 40.55% (RBM), 29.68% (RSM), and 58.9% (RCM) of the variation found in the markers (Figure 6). These results were higher than those reported in similar studies [40,41]. These authors found that 10.9% and 8.79%, respectively, of the variation was explained by the first two dimensions.

For traits RM and RSM, which possess a larger number of the loci under study, the number of eigenvalues considered to reach a minimum of 80% of variability is higher than that for the rest of the traits. However, the first two dimensions possess a high percentage of variation, evident in the graphic representation of the PCA, where the K-means groups are clearly differentiated, particularly for the RCM graph.

For RM, the genes ARHGAP10 and BDH2 (SNPs 2 and 3, respectively) contributed the most to the analysis; however, the relationship between them is negative (Figure 7). This means the genotypes for the genes in an animal are present as AA and in another as BB, or vice versa. Between these genes there was no known linkage or genetic correlation, given that they were in different chromosomes and affect unrelated traits. ARHGAP10 is associated with intramuscular fat formation [56] and BDH2 with feed intake [57].

Oddly enough, for RM, the highly inbred animal 16 is among the top 10% of the most contributory animals (Figure 7). This can probably be explained by its position in the biplot, which means it is most associated with gene TBC1D8, which contributed little in the analysis. The same situation is true for SM, animal 143, and the marker for the gene ITPK1.

For SM, the markers for genes ANKRD33B, CTNND2, GRIA3, and SIDT1 contributed the most to the PCA, Figure 6. The pairs of genes CTNND2 with SIDT1, and ANKRD33B with GRIA3 had a positive relationship (Figure 7) even though they are in distant chromosomes. This may be a particular feature of the genotypic profile for mastitis in this Braunvieh population.

The markers of genes IL1A and ITGB3 for RBM were in a relationship similar to that of the genes ARHGAP10 and BDH2 for RM (Figure 6). They had a negative relationship according to the biplot (Figure 7). Similarly, they are in different chromosomes, even though both directly influence the function of the immune system [20,21].

The marker BovineHD0300029925 (10) of HIVEP3 (Figure 6 and Figure 7) contributed the most to the PCA for RSM. This SNP is positively related to another two intragenic HIVEP3 markers: BovineHD0300029904 (9) and BovineHD0300029955 (11). For RCM, most of the markers were highly contributive, i.e., the SNPs for the genes CXCL8, SEL1L, and STAT4, associated directly with the immune system [49].

## 4. Conclusions

The population studied presented a high genetic diversity for resistance and susceptibility to mastitis, even higher than that of the original Braunvieh and Brown Swiss. Although 13.3% were highly inbred animals (>50%), there were highly heterozygous groups in terms of the traits studied, a favorable indicator of the presence of diversity; even though the homozygous genotypes for immunologic genes (group CXCL) might be favorable for traits such as resistance to subclinical mastitis and resistance to clinical mastitis.

The K-means algorithm and the principal components analysis are good statistical tools to visualize groups of animals with different genotypic profiles. The principal components analysis was useful to detect relationships and contributions of markers for each trait. The results of our research give evidence of the genetic potential of the Mexican Braunvieh population for the improvement of mastitis-related traits in the tropic.

## Figures and Tables

**Figure 1 vetsci-08-00329-f001:**
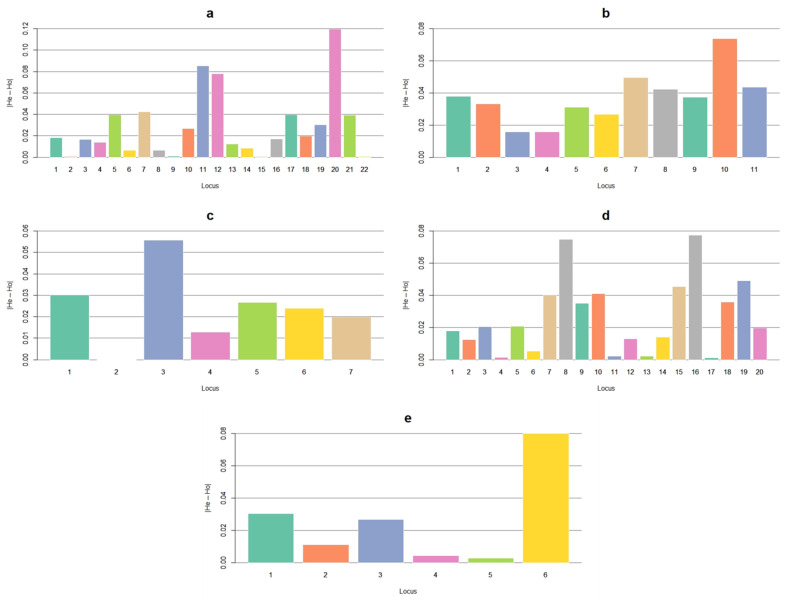
Absolute differences by locus of expected heterozygosity (He) minus observed (Ho) for resistance (**a**) and susceptibility (**b**) to mastitis, resistance to bacterial mastitis (**c**), subclinical mastitis (**d**), and clinical mastitis (**e**) in a Mexican Braunvieh cattle population. |He-Ho| = absolute difference between He and Ho. Note: see the names of each locus by trait in Table 1, Table 2, Table 3, Table 4 and Table 5.

**Figure 2 vetsci-08-00329-f002:**
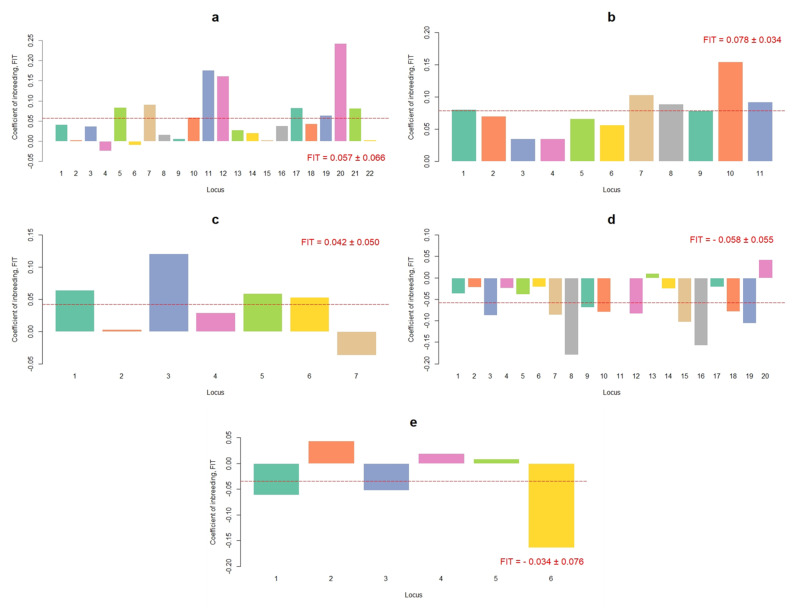
Coefficients of inbreeding (F_IT_) per locus and trait for resistance (**a**) and susceptibility (**b**) to mastitis, resistance to bacterial mastitis (**c**), subclinical mastitis (**d**), and clinical mastitis (**e**) in a Mexican Braunvieh cattle population. The mean of F_IT_ and its standard deviation are shown in red. The red dotted line indicates the mean of F_IT_ for all the loci. Note: see the names of each locus by trait in Table 1, Table 2, Table 3, Table 4 and Table 5.

**Figure 3 vetsci-08-00329-f003:**
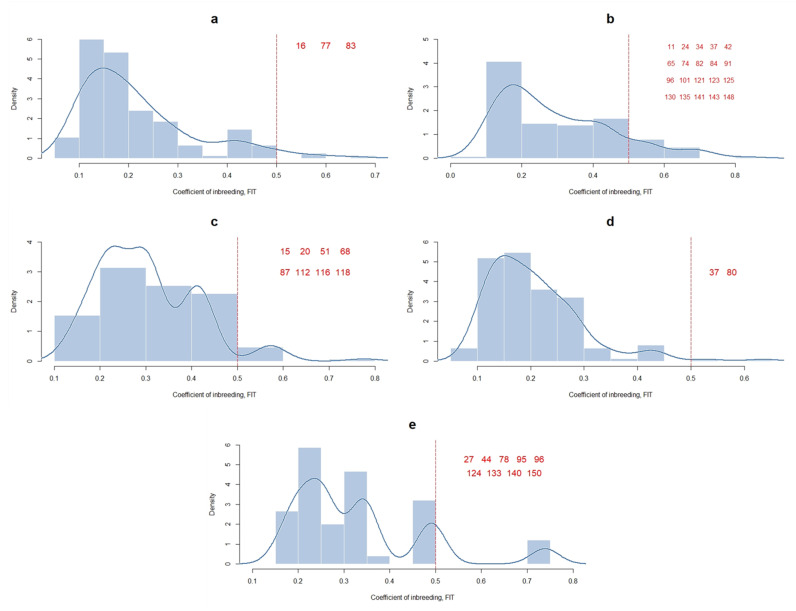
Histogram and density plot (cosine Kernel Function) of the coefficient of inbreeding (F_IT_) for resistance (**a**) and susceptibility (**b**) to mastitis, resistance to bacterial mastitis (**c**), subclinical mastitis (**d**), and clinical mastitis (**e**) in a Mexican Braunvieh cattle population. Bw = bandwidth. Animals with a F_IT_ greater than 0.5 are shown in red on each plot.

**Figure 4 vetsci-08-00329-f004:**
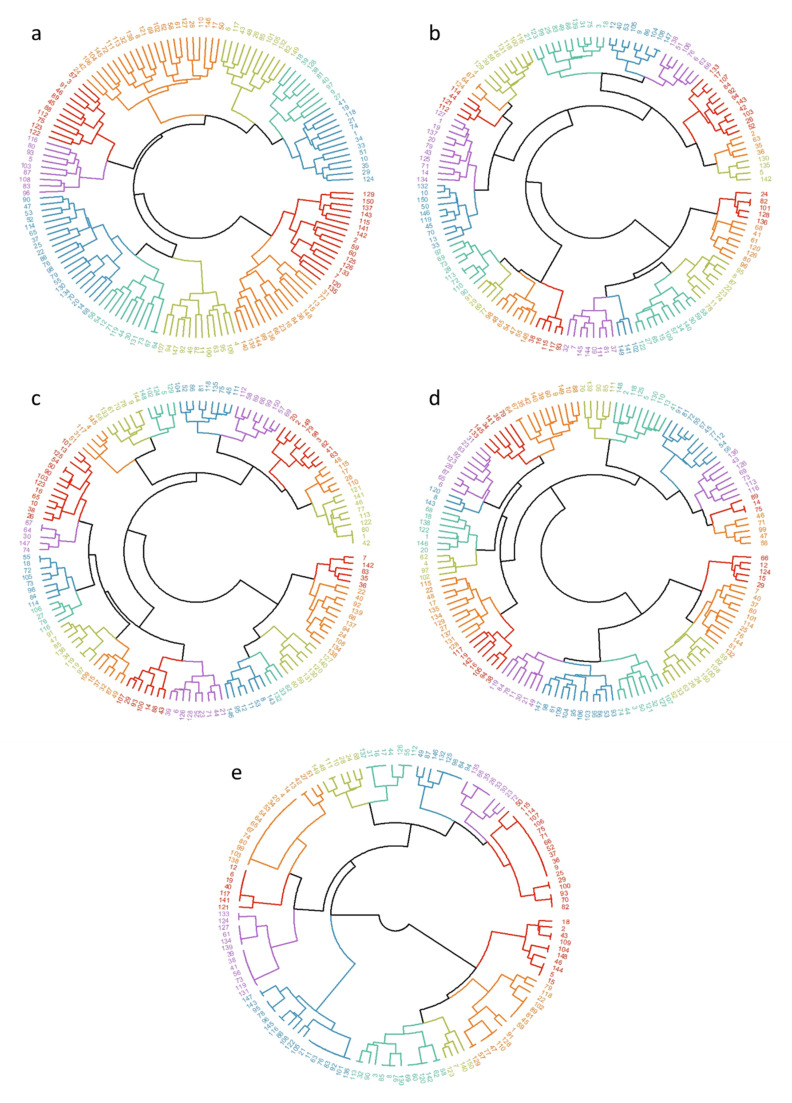
Circular dendrograms obtained by Ward’s algorithm based on the Euclidian distance for resistance (**a**) and susceptibility (**b**) to mastitis, resistance to bacterial mastitis (**c**), subclinical mastitis (**d**), and clinical mastitis (**e**) in a Mexican Braunvieh cattle population.

**Figure 5 vetsci-08-00329-f005:**
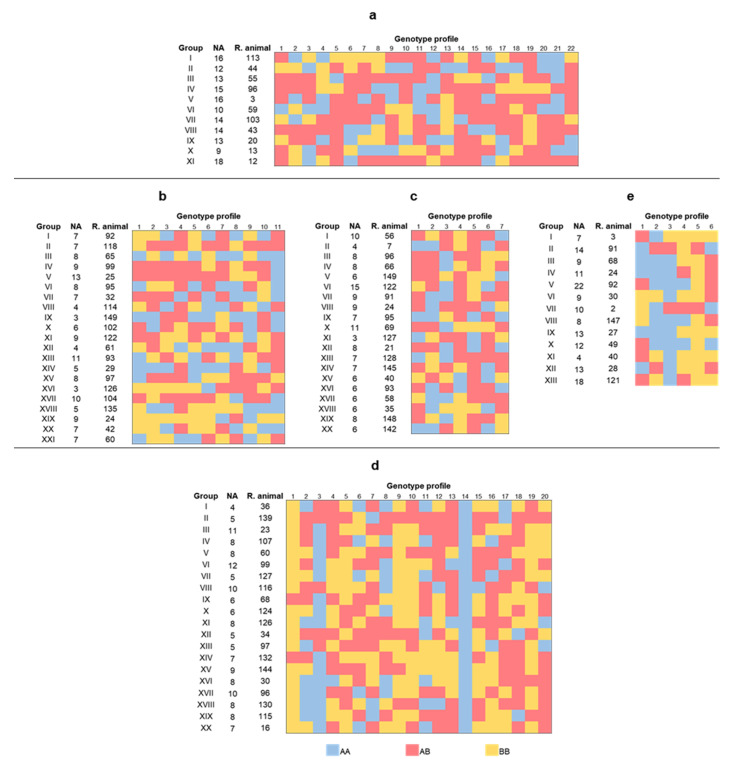
Clustering results for 150 individuals (only one representative animal per group is shown) for resistance (**a**) and susceptibility (**b**) to mastitis, resistance to bacterial mastitis (**c**), subclinical mastitis (**d**), and clinical mastitis (**e**) in a Mexican Braunvieh cattle population. NA = Number of animals in the group. R. animal = representative animal (the closest to the centroid). Note: the names of each locus by trait are indicated in Table 1, Table 2, Table 3, Table 4 and Table 5.

**Figure 6 vetsci-08-00329-f006:**
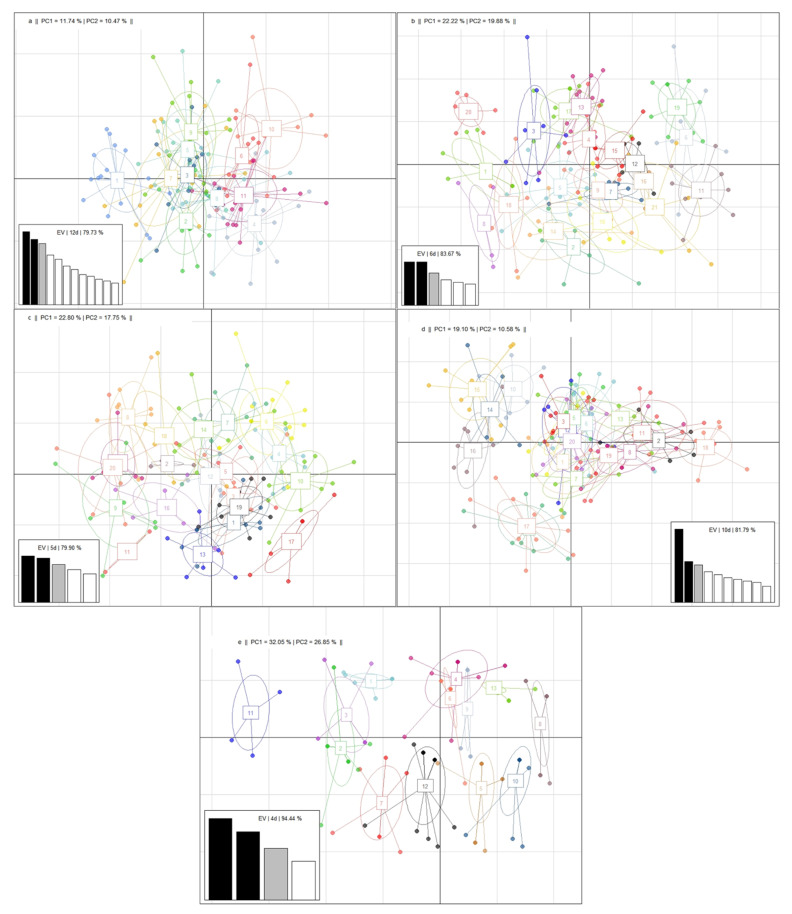
Graphic representation of the principal components analysis: K-means’ grouping of individuals showing eigenvalues (EV) for the dimensions (**d**) whose sum of variability is around 80%, for resistance (**a**) and susceptibility (**b**) to mastitis, resistance to bacterial mastitis (**c**), subclinical mastitis (**d**), and clinical mastitis (**e**) in a Mexican Braunvieh cattle population. PC1 = percentage of variability explained by dimension one. PC2 = percentage of variability explained by dimension two.

**Figure 7 vetsci-08-00329-f007:**
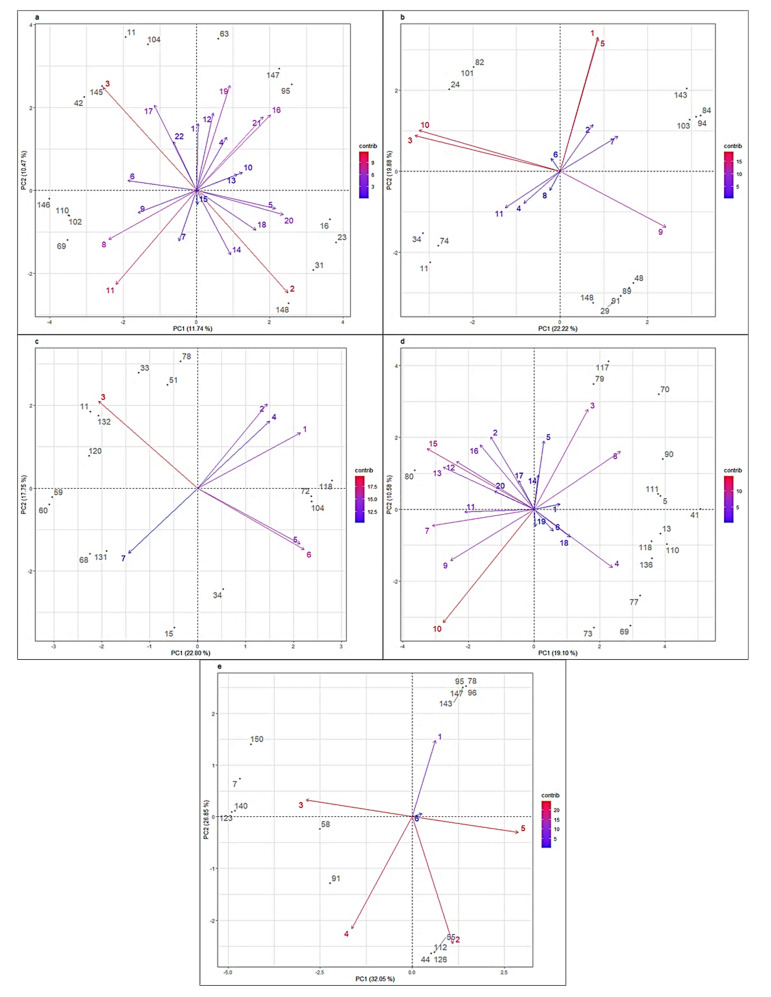
Biplots: marker’s contribution and 10% most contributing individuals for resistance (**a**) and susceptibility (**b**) to mastitis, resistance to bacterial mastitis (**c**), subclinical mastitis (**d**), and clinical mastitis (**e**) in a Mexican Braunvieh cattle population. PC1 = percentage of variability explained by dimension one. PC2 = percentage of variability explained by dimension two. Note: the names of each locus by trait are indicated in Table 1, Table 2, Table 3, Table 4 and Table 5.

**Table 1 vetsci-08-00329-t001:** Top 10% single nucleotide polymorphisms (SNP) associated with resistance to mastitis (RM) in Braunvieh cattle.

Gene	Chr	Position, pb (±25 k)	SNP Marker Name	LN
ARHGAP10 [16]	17	9,994,895–10,429,767	ARS-BFGL-NGS-113821	1
			BovineHD1700002890	2
BDH2 [15]	6	21,737,056–21,815,795	BovineHD0600006001	3
CAPG [14]	11	49,562,863–49,630,835	Hapmap54495-rs29018810	4
CHD5 [14]	16	47,131,943–47,253,618	BovineHD1600013051	5
CST6 [15]	29	44,076,083–44,127,573	BovineHD2900013196	6
ELMO1 [14]	4	59,842,686–60,475,511	BovineHD0400016236	7
FBL [16]	18	49,363,469–49,425,788	ARS-BFGL-NGS-113564	8
FOCAD [14]	8	23,394,437–23,750,228	BovineHD0800007122	9
IMMP2L [14]	4	56,736,178–57,747,698	BovineHD0400015680	10
LOC510112 [16]	10	27,914,007–27,964,951	BovineHD1000009188	11
			BovineHD1000009179	12
MBL2 [17]	26	6,306,933–6,362,539	BovineHD2600001441	13
			BovineHD2600001446	14
MYO1E [14]	10	50,932,729–51,205,381	BovineHD1000015285	15
PON1 [14]	4	12,517,349–12,601,241	ARS-BFGL-NGS-48351	16
PPP3CA [14]	6	23,419,690–23,795,287	BovineHD0600006555	17
ST7 [14]	4	51,162,819–51,481,394	BovineHD0400014185	18
SULF2 [15]	13	76,130,676–76,307,180	BovineHD1300022072	19
TBC1D8 [14]	11	5,953,170–6,144,340	BovineHD1100002208	20
TBCK [14]	6	18,840,254–19,099,244	BovineHD0600005226	21
ZNFX1 [15]	13	77,323,734–77,400,021	BovineHD1300022388	22

Chr = chromosome. Position, pb (±25 k) = candidate gene position in base pairs reported in [26], for *Bos taurus* ± 25,000 base pairs. LN = locus number assigned in graphs.

**Table 2 vetsci-08-00329-t002:** Top 10% single nucleotide polymorphisms (SNP) associated with susceptibility to mastitis (SM) in Braunvieh cattle.

Gene	Chr	Position, pb (±25 k)	SNP Marker Name	LN
ANKRD33B [18]	20	62,576,901–62,728,565	ARS-BFGL-NGS-112060	1
CTNND2 [18]	20	61,189,491–62,339,843	BovineHD2000017160	2
			BovineHD2000017153	3
			BTB-00791947	4
GRIA3 [19]	10	7,542,786–7,901,396	ARS-BFGL-NGS-87466	5
ILDR2 [19]	3	1,878,471–1,997,329	Hapmap42630-BTA-67480	6
ITPK1 [19]	21	57,564,349–57,778,522	BovineHD2100016547	7
			BovineHD2100016592	8
			BovineHD2100016552	9
SIDT1 [19]	1	58,193,233–58,355,060	BovineHD0100016494	10
TBXAS1 [19]	4	103,270,503–103,494,001	BovineHD0400029111	11

Chr = chromosome. Position, pb (±25 k) = candidate gene position in base pairs reported in [26] for *Bos taurus* ± 25,000 base pairs. LN = locus number assigned in graphs.

**Table 3 vetsci-08-00329-t003:** Top 10% single nucleotide polymorphisms (SNP) associated with resistance to bacterial mastitis (RBM) in Braunvieh cattle.

Gene	Chr	Position, pb (±25 k)	SNP Marker Name	LN
ECHDC1 ^1^ [22]	9	23,922,553–24,025,911	BovineHD0900006460	1
GNAI1 ^2^ [20]	4	40,930,829–41,086,584	ARS-BFGL-NGS-110306	2
IL1A ^3^ [21]	11	46,457,553–46,518,533	BovineHD1100013546	3
IL1RN ^2,3^ [20]	11	46,790,914–46,862,407	ARS-BFGL-NGS-113289	4
			ARS-BFGL-NGS-113289	4
ITGA4 ^2^ [20]	2	15,058,580–15,200,364	BovineHD0200004269	5
ITGB3 ^2^ [20]	19	46,305,531–46,418,474	UA-IFASA-8333	6
NRG1 ^1^ [22]	27	28,396,076–28,680,095	BovineHD2700007962	7

Superscripts indicate resistance to ^1^ *S. aureus*, ^2^ *S. agalactiae,* and ^3^ *E. coli*. Chr = chromosome. Position, pb (±25 k) = candidate gene position in base pairs reported in [26] for *Bos taurus* ± 25,000 base pairs. LN = locus number assigned in graphs.

**Table 4 vetsci-08-00329-t004:** Top 10% single nucleotide polymorphisms (SNP) associated with resistance to subclinic mastitis (RSM) in Braunvieh cattle.

Gene	Chr	Position, pb (±25 k)	SNP Marker Name	LN
BMPR1B	6	29,346,363–29,845,366	BovineHD0600008191	1
			BovineHD0600008283	2
			BovineHD0600008292	3
EDN2	3	104,654,329–104,730,954	BovineHD0300029984	4
			BovineHD0300029996	5
GUCA2A	3	103,990,493–104,056,213	BTA-98478-no-rs	6
HEYL	3	106,417,713–106,483,224	BTB-00148619	7
HIVEP3	3	104,084,087–104,699,365	ARS-BFGL-NGS-35125	8
			BovineHD0300029904	9
			BovineHD0300029925	10
			BovineHD0300029955	11
			BTB-01612988	12
			Hapmap48983-BTA-100103	13
MACF1	3	106,564,750–106,955,676	ARS-BFGL-NGS-38199	14
			BovineHD0300030641	15
			BovineHD0300030658	16
			BovineHD0300030677	17
			BovineHD0300030702	18
MFSD2A	3	106,097,951–10,616,0784	BovineHD0300030435	19
SH3PXD2A	26	24,136,301–24,435,553	BovineHD2600006239	20

Chr = Chromosome. Position, pb (±25 k) = Candidate gene position in base pairs reported in [26] for *Bos taurus* ± 25,000 base pairs. LN = Locus Number assigned in graphs. Candidate genes reported by [8].

**Table 5 vetsci-08-00329-t005:** Top 10% single nucleotide polymorphisms (SNP) associated with Resistance to Clinical Mastitis (RCM) in Braunvieh Cattle.

Gene	Chr	Position, pb (±25 k)	SNP Marker Name	LN
CXCL1	6	89,047,989–89,100,128	BovineHD0600024410	1
CXCL8	6	88,785,817–88,839,572	ARS-BFGL-NGS-17376	2
			BovineHD0600024315	3
			BovineHD0600024328	4
SEL1L	10	92,733,415–92,848,117	BovineHD1000026808	5
STAT4	2	79,543,834–79,730,325	BovineHD0200022927	6

Chr = chromosome. Position, pb (±25 k) = candidate gene position in base pairs reported in [26] for *Bos taurus* ± 25,000 base pairs. LN = locus number assigned in graphs. Candidate genes reported by [23].

**Table 6 vetsci-08-00329-t006:** Heterozygosity and *t*-test results by trait associated with mastitis, for a Mexican Braunvieh Cattle population.

Trait	He ± SD	Ho ± SD	NL	*p*-Value BC	*p*-Value *t*-Test	He-Ho
Resistance to Mastitis	0.495 ± 0.006	0.468 ± 0.034	22	4.5 × 10^−4^	5.5 × 10^−4^	0.027
Susceptibility to Mastitis	0.495 ± 0.004	0.458 ± 0.019	11	9.1 × 10^−4^	1.0 × 10^−5^	0.037 *
Resistance to Bacterial Mastitis	0.488 ± 0.010	0.470 ± 0.031	7	1.4 × 10^−3^	4.4 × 10^−2^	0.018
Resistance to Subclinical Mastitis	0.379 ± 0.153	0.403 ± 0.164	20	5.0 × 10^−4^	9.9 × 10^−1^	−0.024
Resistance to Clinical Mastitis	0.406 ± 0.105	0.426 ± 0.130	6	1.7 × 10^−3^	8.9 × 10^−1^	−0.020

He ± SD = Heterozygosity expected ± standard deviation. Ho ± SD = Heterozygosity observed ± standard deviation. NL = number of loci. *p*-value BC = *p*-value with Bonferroni correction (0.01/NL). *p*-value *t*-test = *p*-value obtained by the *t*-test, where the null hypothesis established that the differences between, He and Ho from zero were non-significant, at 0.01/NL significance level. He-Ho = Differences between heterozygosity values (expected-observed), values * indicate statistically significant difference.

## Data Availability

Upon request to the author for correspondence.

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
