# Peer review of "Genetic Diversity and Population Structure for Resistance and Susceptibility to Mastitis in Braunvieh Cattle"

_vetsci, 2021, doi:10.3390/vetsci8120329_

Round 1
Reviewer 1 Report
Introduction
Important issues to be explained, as Veterinary Sciences is not a genetic journal:
Single-nucleotide polymorphism
Quantitative trait locus
Hardy-Weinberg equilibrium
L41 – “factors that influence it and challenge its measurement” – please clarify. (influence mastitis? What mesurements?).
L42 – “… factors to prevent it at…” – replace for “… factors to prevent mastitis at…”
L48 – “The markers most used for…” - replace for “The mostly used markers for…”
L49 – “ single polymorphism nucleotides (SNP).” - replace for “ single nucleotide polymorphism (SNP).”
L59-60 – “… diversity because facing environmental…” - replace for “… diversity as facing environmental…”
Materials and Methods
Explain what you mean for:
1 - resistance to mastitis (RM)
2 - resistance to bacterial mastitis (RBM)
3 - resistance to subclinical mastitis (RSM)
4 - resistance to clinical mastitis (RCM)
The first comprises the other three…
Also, susceptibility to mastitis is merely the opposite to resistance. So, the same loci must be involved…
What was the rationale for distributing the locus in the 5 groups?
Results and Discussion
L257 – The He-Ho values on Table 6 for RSM and RCM don’t seem to match the values on Figure 1 d) and e).
L282 and L286 – “RS” - replace for “SM”
L306-L309 – “RM is multifactorial, and therefore, it includes candidate genes associated with other productive, reproductive, and adaptative traits. This will cause high genetic diversity, in contrast with the traits RSM and RCM, which are influenced by immunological candidate genes”
Please clarify this idea. What do you mean by RM being multifactorial? Aren’t RSM and RCM also multifactorial? Indeed, immunological genes affect resistance to mastitis. Both clinical and subclinical…
L334 – “22.21% to 58.9%” – Are these average values? Please clarify.
The author states in the conclusion
“The Principal Components Analysis was useful to detect relationships and contributions of markers for each trait.”
This was not that clear to me in the Results and Discussion. I think the discussion could be improved.
Author Response
- Important issues to be explained, as Veterinary Sciences is not a genetic journal:
Thanks for the comment, we are sure what follows will help the reader to understand the details of the research.
One of our objectives was to tackle the disease mastitis from a molecular perspective. Additionally, today one way or another all the areas of animal production/science are related with molecular advances. So we think our research in not completely focused only on the molecular side.
Single-nucleotide polymorphism: we give a small definition (L55-L58); we think that the term SNP is widely known today so it would not need a very detailed explanation.
Quantitative trait locus: we define the term QTL and its implications in our study (L35-L40).
Hardy-Weinberg equilibrium: we include a complete paragraph to describe some parameters and tools used in genetic diversity studies, we also include the definition of Hardy-Weinberg equilibrium (L71-L77).
- Language and mistakes
We appreciate the time it took to read our manuscript in detail. All suggested corrections were made.
L41 – “factors that influence it and challenge its measurement” – please clarify. (Influence mastitis? What measurements?)
You are right, the idea was not clear. We mean that resistance to mastitis is a difficult characteristic to be measured. We corrected the wording (L46-L48).
L42 – “… factors to prevent it at…” – replace for “… factors to prevent mastitis at…”
We corrected the wording (L49).
L48 – “The markers most used for…” - replace for “The mostly used markers for…”
We corrected the wording (L55).
L49 – “ single polymorphism nucleotides (SNP).” - replace for “ single nucleotide polymorphism (SNP).”
We regret this evident mistake. It was corrected (L56).
L59-60 – “… diversity because facing environmental…” - replace for “… diversity as facing environmental…”67
We corrected the wording (L68-L69).
Materials and Methods
- Explain what you mean for:
1 - resistance to mastitis (RM)
2 - resistance to bacterial mastitis (RBM)
3 - resistance to subclinical mastitis (RSM)
4 - resistance to clinical mastitis (RCM)
The first comprises the other three…
Also, susceptibility to mastitis is merely the opposite to resistance. So, the same loci must be involved…
What was the rationale for distributing the locus in the 5 groups?
Thanks for this observation which is absolutely pertinent.
Of course, resistance to mastitis comprises all three traits; however, this does not mean that exactly the same loci are involved for all traits. Something similar happens with RM and SM. Therefore, the different studies that found the candidate genes for the mastitis traits used in this study, report different loci results for each trait. In the article, we give a concrete explanation of why the candidate genes for each trait could be different (L104-L114). Additionally, we filtered the markers using the Shannon index, this may have eliminated some common markers for the traits. We report the 10% of the most informative loci by trait, perhaps some of the least informative loci were the same for all traits, and we discarded them in the genotypes edition.
The main reason for studying the five traits is that we try to determine the differences in genetic diversity and population structure when studying resistance to different types of mastitis (L106-L08). As described in the results, there are indeed differences between traits for the inbreeding coefficient and for diversity in general.
Results and Discussion
- L257 – The He-Ho values on Table 6 for RSM and RCM don’t seem to match the values on Figure 1 d) and e).
They do not agree since in the table they are given their real value (with a negative sign) and in the barplot, we represent it in absolute numbers (positive). We prioritize quality in observing the difference of Ho and He in the graph, the comparison between the sizes of the bars is clearer when they are all in the same quadrant. In the description of Figure 1, it is explained that they are absolute values or | He-Ho |. (L313-L316).
- L282 and L286 – “RS” - replace for “SM”
We regret this evident error (L333).
- L306-L309 – “RM is multifactorial, and therefore, it includes candidate genes associated with other productive, reproductive, and adaptative traits. This will cause high genetic diversity, in contrast with the traits RSM and RCM, which are influenced by immunological candidate genes”
Please clarify this idea. What do you mean by RM being multifactorial? Aren’t RSM and RCM also multifactorial? Indeed, immunological genes affect resistance to mastitis. Both clinical and subclinical…
Thanks for this observation. Obviously, it was not as clear as we expected. We have already corrected it in the manuscript. We wanted to say that resistance to mastitis may have more candidate genes associated with non-immunological factors than the other traits. Furthermore, specifically for our study resistance to subclinical and clinical mastitis are characterized mostly by genes that affect immunological processes. Hopefully the idea is now clearer (L358-L361).
- L334 – “22.21% to 58.9%” – Are these average values? Please clarify.
Thanks, we think the expression will be better if we described the variation found by trait. As we did (L378-L379).
- The author states in the conclusion
“The Principal Components Analysis was useful to detect relationships and contributions of markers for each trait.”
This was not that clear to me in the Results and Discussion. I think the discussion could be improved.
We really appreciate your suggestion; however, we think that the statements in L-378-L379, L388-L390, L-394-L397, and L425-L427 give enough evidence to support our conclusion. As we wrote in the manuscript, we obtained a high variation explained by the dimensions of PCA (22.21%, 42.1%, 40.55%, 29.68%, and 58.9%), so we think our results are conclusive thanks to PCA. With PCA we could see the distribution of the animal's groups and visualize the K-means results. Additionally, we discussed the most important correlations between the markers and markers contributions, this was only possible thanks to the PCA and biplot graphics and results.

Reviewer 2 Report
(1) The chip of the 70 Genomic Profile Bovine LD with 50,000 SNP markers were used to study Mexico Braunvieh cattle for resistance to mastitis (RM), susceptibility to mastitis (SM), resistance to bacterial mastitis (RBM), resistance to subclinical mastitis (RSM) and resistance to clinical mastitis (RCM).
(2) What’s the difference of candidate genes between this study and your previous study or other research groups’ studies? Why?
(3) In the part of conclusion, you deduce group CXC might be favorable for traits such as resistance to subclinical mastitis and resistance to clinical mastitis, can you give us solid evidences (in vivo or in vitro) to confirm this conclusion.
Author Response
The chip of the Genomic Profile Bovine LD with 50,000 SNP markers were used to study Mexico Braunvieh cattle for resistance to mastitis (RM), susceptibility to mastitis (SM), resistance to bacterial mastitis (RBM), resistance to subclinical mastitis (RSM) and resistance to clinical mastitis (RCM).
(2) What’s the difference of candidate genes between this study and your previous study or other research groups’ studies? Why?
Thanks for these comments, we think could be good to explain it. Actually, we did it in Lines 189-191. Where we explain the candidate genes used in each study. In summary, in our project, we have tried to determine the genetic diversity of our Braunvieh population for different traits, as a first step in the search to improve milk and meat production in dual-purpose systems.
(3) In the part of conclusion, you deduce group CXC might be favorable for traits such as resistance to subclinical mastitis and resistance to clinical mastitis, can you give us solid evidences (in vivo or in vitro) to confirm this conclusion 254-256
We really appreciate this recommendation; it will make our conclusion stronger. We found the required evidence; we explain that in an in vivo study, Kelly-Scumpia et al. (2010) found that CXCL genes are key in the inflammatory process of the mammary gland. You can see this in Lines 287-290 of the manuscript.

Reviewer 3 Report
Dear authors
your study is ready for publishing only some minor changes I needed. I send you my minor remarks.

Author Response
Thanks for your valuable comments, they will definitely make our manuscript wording better. Here you can find the corrections:
L12, we change “…to improve…” to “…to be improved…”
L34, we change “…to measure…” to ”…to be measured…”
L42, we changed “…to improve…” to “…to be improved…”

Round 2
Reviewer 2 Report
Accepted for publication.